# Clinical Value of Muscle Mass Assessment in Clinical Conditions Associated with Malnutrition

**DOI:** 10.3390/jcm8071040

**Published:** 2019-07-17

**Authors:** Julie Mareschal, Najate Achamrah, Kristina Norman, Laurence Genton

**Affiliations:** 1Clinical Nutrition, Geneva University Hospitals, 1205 Geneva, Switzerland; 2Department of Clinical Nutrition, Rouen University Hospital, Normandie University, 76000 Rouen, France; 3Research Group on Geriatrics, Charité Universitätsmedizin Berlin, Corporate Member of Freie Universität Berlin, Humboldt-Universität zu Berlin, and Berlin Institute of Health, 13347 Berlin, Germany; 4Department of Nutrition and Gerontology, German Institute for Human Nutrition Potsdam-Rehbrücke, 14558 Nuthetal, Germany; 5Clinical Nutrition, Geneva University Hospital, 1205 Geneva, Switzerland

**Keywords:** mid-arm muscle circumference, bioelectrical impedance analysis, dual-energy X-ray absorptiometry, computed tomography, fat-free mass, appendicular skeletal muscle mass, lean soft tissue, skeletal muscle index, chronic disease, old

## Abstract

Malnutrition results from a reduction of food intake or an alteration of nutrient assimilation and leads to decreased lean mass. Strong evidence shows that malnutrition associated with loss of muscle mass negatively impacts clinical outcomes. The preservation or improvement of muscle mass represents a challenge. This review aims to (1) describe current methods to assess muscle mass in clinical practice, (2) describe the associations between muscle mass and clinical outcomes, and (3) describe the impact of interventions aiming at increasing muscle mass on clinical outcomes. It highlights the importance of assessing muscle mass as part of the screening and the follow-up of malnutrition in clinical practice.

## 1. Introduction

Malnutrition results from a reduction of food intake or alteration of nutrient assimilation and leads to decreased lean mass, either combined or not with the loss of fat mass. Potential causes include diseases, starvation, or aging. This condition may be associated with other homeostasis disorders such as inflammation [1]. Malnutrition negatively impacts clinical outcomes, mortality, length of stay, and costs [2]. In hospitalized patients, the prevalence ranges from 20 to 50% [3].

To standardize the definition of malnutrition and its diagnostic criteria, the Global Leadership Initiative on Malnutrition (GLIM) has recently convened experts of major worldwide clinical nutrition societies. They suggested defining malnutrition with one phenotypic criterion (bodyweight loss, low body mass index (BMI), or reduced muscle mass) associated to one etiologic criterion (reduced food intake/assimilation or inflammation/disease burden) [4]. Thus, the experts promote the use of body composition measurement as part of a nutritional assessment to evaluate muscle mass. 

This narrative review aims to (1) describe current methods to assess muscle mass in clinical practice, (2) describe the associations between muscle mass and clinical outcomes, and (3) describe the impact of interventions aiming at increasing muscle mass on clinical outcomes. 

## 2. Methods to Assess Muscle Mass in Clinical Practice

In clinical practice, several tools and techniques are available to assess body composition. BMI is often used due to its simplicity. Indeed, the U-shaped association between BMI and all-cause mortality has been well described, as subjects with the highest and lowest BMIs have the highest mortality rates [5]. However, BMI does not allow the measurement of body composition compartments and tends to underestimate fat-free mass (FFM) depletion [6,7,8]. Other anthropometric measurements such as skinfold thickness and waist-hip ratio can also be used. Even though these tools are convenient, quick, and inexpensive, they do not provide direct information on muscle mass [9,10]. Therefore, other methods are required to assess muscle mass in clinical practice. In this section, we focused on portable techniques which can be used at patient bedsides, such as mid-arm muscle circumference and bioelectrical impedance analysis. We concentrated also on dual-energy X-ray absorptiometry and computed tomography, non-portable techniques used for other clinical diagnostic purposes, but allows for the simultaneous assessment of body composition. The characteristics of each method are compared in Table 1. Other methods to assess muscle mass were not introduced because they are rarely used in clinical practice.

### 2.1. Mid-Arm Muscle Circumference

Mid-arm muscle circumference (MAMC) is obtained by using the formula: MAMC (mm) = mid-arm circumference (mm) − (3.14 × triceps skinfold(mm)) [12]. Measurements are usually performed in standing position, on the dominant arm, at the mid-point between the acromion and the olecranon. Mid-arm circumference is measured using a plastic metric tape and triceps skinfold using a skinfold caliper. For both parameters, the average of three consecutive measurements is recorded [12]. MAMC provides an estimation of upper extremity skeletal muscle mass, which strongly correlates to dual-energy X-ray absorptiometry results [13,14].

This method is quick, portable, inexpensive, and easy to perform. It requires simple equipment, minimal training, and is useful in patients with ascites and edema. The main disadvantages are interobserver variations [15]. For instance, in obese people, accuracy is low due to the difficulty of the required triceps skinfold measurement [16]. 

### 2.2. Bioelectrical Impedance Analysis

Several bioelectrical impedance analysis (BIA) methods are available (single frequency, multi-frequency, segmental, or vector BIA). We focused on single-frequency 50 kHz tetrapolar BIA, as this is the most used method in clinical practice, and on BIA devices with hand-to-foot surface electrodes, as these BIA devices provide raw electrical data. BIA allows to obtain whole FFM, which refers to all body compartments except fat mass, and appendicular skeletal muscle mass (ASMM) defined as the sum of the lean soft tissue of the four limbs. 

Principles and methods of BIA have been previously detailed in the guidelines of the European Society for Clinical Nutrition and Metabolism [17]. BIA is based on the concept that adipose tissue is more resistant to the conduction of the current compared to other tissues and fluids. Briefly, the patient, lying on the back, is exposed to a low-intensity alternating current between surface electrodes. The BIA device measures resistance and reactance, or impedance and phase angle. To estimate FFM and ASMM, several population-specific equations using these electrical parameters in addition to age, sex, weight, and height have been developed and validated against dual-energy X-ray absorptiometry (DXA) [18,19,20,21]. FFM and ASMM can be divided by height squared to be converted into FFM index (FFMI) and ASMM index (ASMMI), in order to compare people of different heights.

As a portable non-invasive bedside method, BIA is convenient in clinical practice. It is safe, cheap and not technically demanding. The small interobserver variability, 0.02 kg for FFM, is an advantage compared to anthropometry methods [22,23]. This device also allows the calculation of phase angle, supposed to be an indicator of cellular integrity, associated with clinical prognosis [24]. FFM assessment using BIA does, however, bear some limitations. BIA is not accurate in patients with altered hydration (e.g., ascites, edema, fluid loss) and with extreme BMI (<16 kg/m^2^ and >34 kg/m^2^) [25]. BIA formulas are population-specific. Finally, multiple devices of BIA are commercially available, but integrated algorithms are not always released by the manufacturer, thus questioning the reliability of the results measured by these devices [26,27]. 

### 2.3. Dual-Energy X-ray Absorptiometry

Dual-energy X-ray absorptiometry (DXA) allows the measurement of bone, lean soft tissue, and fat mass. Compared to FFM, lean soft tissue contains essential and non-essential lipids, but not bone mass. DXA device uses X-rays of low- and high-photon energy. As they cross the body, they are attenuated according to the composition and thickness of the encountered body tissues. A detector on the opposite side of the body analyzes the transmitted photon intensity. Complex algorithms allow then to differentiate bone, fat mass, and lean soft tissue, as detailed previously [28]. This method, initially used for bone density measurement, is often considered as the reference method to assess body composition in clinical research [29]. 

In clinical practice, DXA allows accurate, fast, and non-invasive lean soft tissue assessment. This device is often available in developed countries. Measurement is achieved in supine position. Irradiation is acceptable (2–5 µSv), which is low compared to the daily background radiation of 5–7 µSv [28]. Besides whole-body composition, DXA also allows for the measurement of ASMM. As DXA calculations are based on a constant hydration of lean soft tissue, hydration level variations may impact the results. However, this effect seems to be negligible [30]. Finally, measurements may be influenced by the thickness of the tissue with potential underestimation of fat mass and overestimation of muscle mass in obese patients [31]. 

### 2.4. Computed Tomography

Computed tomography (CT) to assess muscle mass is becoming more common [32]. As cutoff values have been defined at the third lumbar vertebrae (L3) level, an area reflective of whole-body tissue distribution, we focused only on muscle assessment at this anatomic landmark [33,34]. L3 muscle mass area is usually normalized to the patients’ height squared to determine skeletal muscle index (SMI).

CT is a medical imaging technique, performed in supine position, which measures tissue absorption of X-rays emitted through a rotating beam. Computer processing then reconstructs cross-sectional images of anatomical structures by 2D and 3D maps of pixels. According to the attenuation of the different tissues, each pixel is associated with a numerical value (Hounsfield Unit). Tissues are identified in the cross-sectional images by their specific absorption Hounsfield Unit ranges [35,36]. Muscle mass area is obtained by cross-sectional analysis using standard radiology software [37]. 

CT provides high-quality images and precise assessment of muscle mass [38,39]. Its advantages compared to other methods are to evaluate muscle mass quality by measuring fat infiltration in muscle. CT images are frequently available in cancer and other chronic disease patients, as part of the routine diagnosis or follow-up of these diseases [26]. Major drawbacks are high radiation exposure (10 mSv), cost, non-portability, and need for qualified technicians and specific software. Moreover, a recent systematic review highlighted the lack of consensus and high variability of CT-based methods of muscle mass assessment [32]. 

## 3. Impact of Muscle Mass on Clinical Outcomes

The impact of muscle mass on clinical outcomes is well described in different populations. In this section, we focused on chronic diseases such as chronic obstructive pulmonary disease (COPD), chronic heart failure (CHF), cancer, and on older adults. This choice was related to the fact that the prevalence of malnutrition is particularly high in these populations [40,41,42]. We made an update of the latest observational studies and subjectively considered articles from the last three years. 

### 3.1. Chronic Diseases 

The prevalence of malnutrition is significant in patients with chronic diseases, such as for instance, COPD, CHF or solid tumors cancer. It ranges respectively from 20–35%, 60–69% and 14–66% according to the tumor site [43,44,45]. 

#### 3.1.1. Chronic Obstructive Pulmonary Disease 

In patients with COPD, the influence of muscle mass on the severity of the disease and the prognosis has been described by Munhoz et al. [46]. In this study, disease severity was evaluated with the Global Initiative for Obstructive Lung Disease (GOLD) index based on airflow limitation, exacerbation history, symptom burden, and prognosis with the body mass index/airflow obstruction/dyspnea/exercise capacity score (BODE), known to predict disease severity and mortality. Interestingly, ASMMI assessed by DXA decreased significantly according to the worsening of GOLD and BODE indexes. In another study, Matkovic et al. have been interested in the association between body composition and physical performance [47]. In 111 moderate to very severe COPD patients, FFMI, assessed by DXA, were significantly associated with a low capacity exercise and physical activity defined, respectively, by a 6-minute walk distance ≤350 m and a daily step count ≤7128 steps/day (=median). Finally, emphysema severity in COPD patients, characterized by a loss of lung tissue, seems to be related to muscle mass and prognosis [48]. Patients in the higher quartiles of emphysema severity had lower FFMI, evaluated by BIA, and a worse BODE index. Thus, in patients with COPD, muscle mass is associated with disease severity, prognosis, and physical performance.

#### 3.1.2. Chronic Heart Failure

As for patients with COPD, muscle mass also appears to be a good predictor of exercise capacity in patients with chronic heart failure [49]. In 117 patients with heart failure and preserved left ventricular ejection fraction, ASMM measured by DXA was significantly associated with a 6-minute walk test <400 m. Tsuchida et al. have studied the association between muscle mass obtained by DXA and severity of acute decompensated heart failure characterized by brain natriuretic peptide (BNP) > 500 pg/mL [50]. Low ASMMI was defined as two standard deviations below the mean reference values of healthy Japanese subjects [51]. After adjustment for anemia and atrial fibrillation, a low ASMMI was related to a higher BNP level, indicative of poor prognosis in CHF patients [52]. In patients with heart failure, muscle mass is associated with physical performance, severity of the disease, and prognosis.

#### 3.1.3. Cancer 

Figure 1 highlights the association between muscle mass and clinical outcomes in patients with cancer. In diverse solid tumor types, muscle mass is related with mortality, surgical complications, and quality of life. For advanced cell lung cancer, an association between muscle mass and physical function has been reported. Interestingly, early chemotherapy discontinuation and delayed chemotherapy also appear to be related to the amount of muscle mass. In summary, maintaining muscle mass is essential in cancer patients to improve overall survival, quality of life, physical exercise capacity, tolerance to cancer treatments, and to decrease postoperative mortality as well as complications.

### 3.2. Older Adults

Due to aging, the risk of muscle mass depletion is high in older adults. In this population, malnutrition ranges from 29% to 61% according to the diagnostic criteria [40]. Recently, muscle mass has been included in the definition of sarcopenia published by the European Working Group on Sarcopenia in Older People [72]. Over the last three years, we found over 80 studies dealing with muscle mass in older adults. Due to the large amount of publications, Figure 2 considers studies including more than 100 participants and illustrates the effect of low muscle mass according to the clinical setting. Studies including fewer participants showed a similar impact. 

## 4. Improvement in Muscle Mass: Strategies and Clinical Benefits

Considering the negative effects of muscle mass loss, preserving or increasing muscle mass could lead to improvement of clinical outcomes. Therapeutic strategies to achieve this goal may include nutritional intervention, physical exercise, anabolic steroids, and growth hormone. Nutritional support is recommended for every malnourished patient, as defined by the GLIM [4]. Regular physical exercise is promoted for patients with COPD, chronic heart failure, and cancer as it improves cardiorespiratory fitness, muscle mass and strength, quality of life, and decreases COPD exacerbation and chemotherapy toxicity [81,82,83,84]. Anabolic steroids and growth hormone have been considered in malnourished patients, but no clinical practice guideline has been published yet [85,86]. These strategies are used either individually or as a multimodal treatment in clinical research with the aim to prevent muscle mass loss. In this section, we focused on non-pilot randomized controlled trials published during the last three years. We did not find new relevant studies with growth hormone supplementation. 

### 4.1. Chronic Diseases 

#### 4.1.1. Chronic Obstructive Pulmonary Disease 

Calder et al. evaluated the benefits of a 12-week specific oral nutritional supplementation (~230 kcal, 10 g whey proteins, enriched with omega 3 and vitamin D) vs. milk comparator (~200 kcal, 10 g proteins) in moderate to severe COPD with a BMI between 16–18 kg/m^2^ and involuntary weight loss [87]. Although improvement in dyspnea was demonstrated in the intervention group, no modification of muscle mass was observed in either group. In another study, Van de Bool et al. demonstrated the interest of a 4-month multimodal rehabilitation, including nutritional supplementation and physical activity in moderate airflow limitation COPD patients with low muscle mass [88]. Low muscle mass was defined as a lean soft tissue index measured by DXA, under the sex and age-specific 25th percentile values published by Schutz et al. [89]. The intervention group consumed each day two or three oral nutritional supplements enriched in leucine, omega 3 and vitamin D (1 unit = 187.5 kcal, 9.4 g proteins) and underwent a supervised endurance/resistance training two to three times a week. Patients in the control group were only assigned to a supervised exercise program. In both groups, improvement in ASMM, quadriceps muscle strength, and endurance performance were observed. Inspiratory muscle strength, physical activity level, plasma vitamin D, eicosapentaenoic, and docosahexaenoic acids were improved only in the intervention group. 

#### 4.1.2. Chronic Heart Failure 

Dos Santos et al. randomized CHF patients with testosterone deficiency in a 4-month exercise program, testosterone injection, or combined exercise program and testosterone injection groups [90]. The exercise program consisted of 60 min sessions, three times a week, with stretching, endurance and resistance exercises. Patients with testosterone injection received one testosterone intramuscular injection (1000 mg of testosterone undecyclate) at the beginning of the study. Lean mass was assessed by DXA before and at the end of the intervention. The exercise program, isolated or combined with testosterone injection, increased significantly lean mass (*p* < 0.01) while testosterone injection alone was associated with decreased lean mass (*p* < 0.01). Nutritional intake has not been evaluated.

#### 4.1.3. Cancer 

Randomized controlled trials studying the effects of diverse interventions on muscle mass are presented in Table 2. Only randomized controlled trials with nutritional or physical exercise interventions were found. Results are heterogeneous, probably due to significant differences in types of intervention and population. However, most studies show an increase in muscle mass and other outcomes such as muscle strength. 

Under different conditions, interventions such as nutrition, physical exercise, and anabolic steroids are efficient to prevent the decrease of muscle mass and improve functional and biological parameters. In clinical practice, a body composition assessment should be used to monitor the effects of these interventions.

### 4.2. Older Adults

Table 3 shows randomized controlled trials studying the effects of nutritional or combined nutritional and physical interventions on muscle mass in older adults. To limit the size of the table and facilitate the reading, we reported studies including over 100 participants, but the results were similar in studies with fewer participants. As for cancer patients, population and results are heterogeneous. However, most studies demonstrated positive effects of interventions on physical function but not on muscle mass. In older adults, muscle mass quality and cardiorespiratory capacities could be more essential than muscle mass quantity to improve physical function.

## 5. Use of Body Composition in Clinical Practice

Current trends towards the aging population and increased prevalence of chronic diseases will continue to rise in the next decades [100]. Malnutrition will thus likely become more problematic on a large scale and standardized care of this condition is needed. Although convenient and quick, BMI has shown limitations in the screening and the follow-up of malnutrition with a tendency to underestimate muscle mass [6,7,8]. The disparity between BMI and FFM raise the need for a precise quantitative evaluation of muscle mass or muscle function to both direct and validate the effects of clinical interventions in malnourished patients. Indeed, it has been established that a loss of muscle mass is associated with a decrease in physical function or muscle strength [101]. Thus, body composition evaluation should be used for the screening and diagnosis of malnutrition in clinical practice, but also for its follow-up, such as in investigation of weight loss composition following surgery or cancer therapy [4,72,84,102]. Repeated measurements of body composition will allow for the tailoring multimodal therapy. Examples of patient samples are presented in Figure 3. They highlight the clinical importance of body composition assessment to detect changes in muscle mass according to every specific patient event. Furthermore, these examples illustrate the advantage of body composition over BMI. For example, Figure 3a shows a stabilization of FFM but a decrease of total body weight and thus of BMI between July and September 2018. In Figure 3c, BMI tends to decrease as MAMC increases.

To date, MAMC, DXA, BIA, and CT at the L3 level seem to be more relevant to assess body composition in clinical practice. Figure 4 summarizes the best methods to evaluate muscle mass according to techniques availability, patient’s hydration, and BMI. CT is highly precise, but, due to radiation exposure and the existence of less irradiating body composition methods, this technique should only be used in patients who undergo CT scan for other purposes such as diagnosis or follow-up [103]. Of note, deriving body composition from pre-existing clinical images is an opportunity to improve diagnosis and treatment without additional cost or patient burden. Clinical research in this area is warranted. The same applies to DXA, which can be performed without additional radiation, costs, and logistical constraints in patients who already benefit from repetitive measurement of bone density in routine care [11]. These two methods are, however, rarely used for body composition assessment in clinical routine, probably because clinical practitioners and radiologists lack information on these techniques. In other situations, BIA appears to be the most suitable method. Indeed, BIA is a portable non-invasive bedside method, quick, cheap, and reproducible [25]. Finally, MAMC will be useful especially in patients with variations in hydration level (e.g., ascites and edemas) or extreme BMI which are BIA limitations [104]. Furthermore, body composition should be integrated into routine clinical practice for a personalized nutritional support. FFM, including muscle mass, is the primary determinant of resting energy expenditure (REE) [105]. In clinical routine, indirect calorimetry is used to assess REE that reflects vital activities (cardiac, respiratory, secretory, cellular, basal muscle tone). The effect of FFM on REE depends on its quantity, assessed by body composition, and its metabolic activity [106]. Clinical conditions associated with muscle wasting, hypercatabolism and/or immobilization, lead to REE variations. Thus, in clinical practice, combining the measurement of body composition with indirect calorimetry may be useful to optimize the nutritional prescription and for the interpretation of feeding energy needs over time. Finally, clinicians should be trained to routinely use body composition in their practice, and interpreting the results should be included in pre- and post-graduate educational programs, as proposed by the European Society for Clinical Nutrition and Metabolism Life Long Learning (LLL) program. The awareness and the training of clinicians to body composition assessment may be a great opportunity to improve interdisciplinary care in the screening and management of malnutrition.

## 6. Conclusions

Available data suggest that precise assessment of body composition might be clinically relevant in the management of malnourished patients. Various methods have been validated to measure muscle mass. The method selection should be driven by the clinical situation, the patient’s characteristics, and logistic and economic parameters. Standard measures of body composition such as BMI are valuable for their simplicity in daily practice. They do, however, not reflect body composition compartments. There is growing evidence in the literature that repartition of muscle mass is a more valuable tool in this assessment. Knowing this repartition allows a tailored approach to the nutritional treatment of malnourished patients and, according to the literature, led to improved clinical outcomes in various chronic diseases as well as in the older adult. Therefore, there is a need for more systematized data to orientate upcoming clinical guidelines of body composition assessment in malnourished patients.

## Figures and Tables

**Figure 1 jcm-08-01040-f001:**
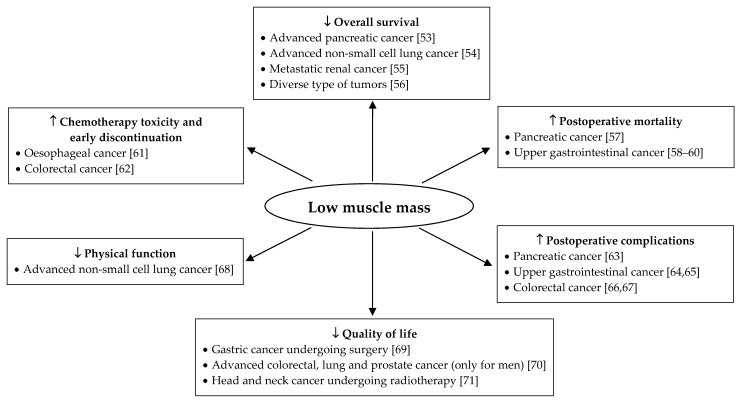
Association between low muscle mass and clinical outcomes in solid tumor cancer patients [53,54,55,56,57,58,59,60,61,62,63,64,65,66,67,68,69,70,71]. Muscle mass was quantified by computed tomography at the L3 level except for references [56] and [71] for which bioelectrical impedance analysis and mid-arm muscle circumference were respectively used.

**Figure 2 jcm-08-01040-f002:**
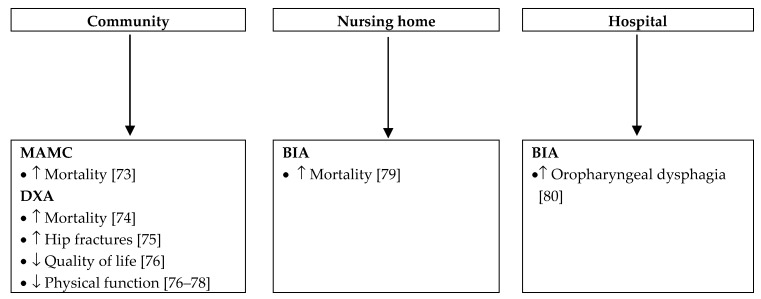
Impact of low muscle mass according to clinical setting [73,74,75,76,77,78,79,80]. MAMC: mid-arm muscle circumference; DXA: dual-energy X-ray absorptiometry; BIA: bioelectrical impedance analysis.

**Figure 3 jcm-08-01040-f003:**
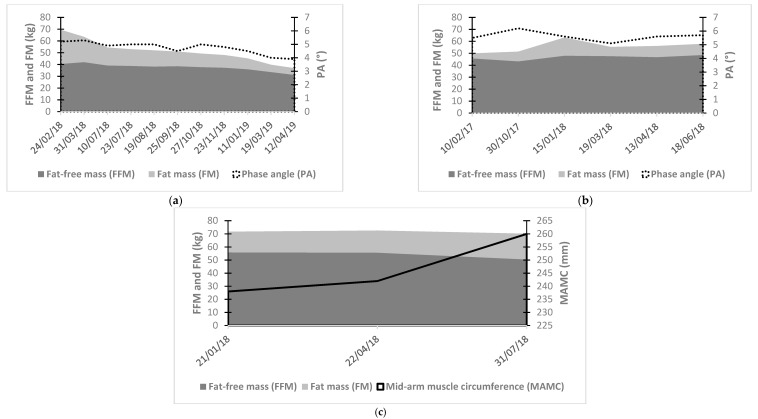
Patients samples (personal data). Evaluation of body composition by 50 kHz bioelectrical impedance analysis: (**a**) Obese patient with gastric cancer. February 2018: total gastrectomy. March to July 2018: severe diarrhea. October 2018: severe nausea. January 2019: tumor recurrence and beginning of a new cycle of chemotherapy until death in April 2019. A decrease in fat-free mass, fat mass and phase angle is observed for each new event and until patient’s death. This example illustrates the association between muscle mass drop and mortality, (**b**) Malnourished COPD patient GOLD stage IV. October 2017: Start of multimodal therapy including enteral support, resistance and endurance physical training and anabolic steroids. An increase of fat-free mass, fat mass and phase angle is observed during the time of multimodal therapy. This example illustrates the importance of body composition assessment to monitor the effects of intervention(s). Evaluation of body composition by 50 kHz bioelectrical impedance analysis and mid-arm muscle circumference: (**c**) Cirrhotic patient with ascites. July 2018: Documented ascites. A decrease of fat-free mass but an increase of mid-arm muscle circumference are observed. This case illustrates BIA limitation in the presence of hydration level variations.

**Figure 4 jcm-08-01040-f004:**
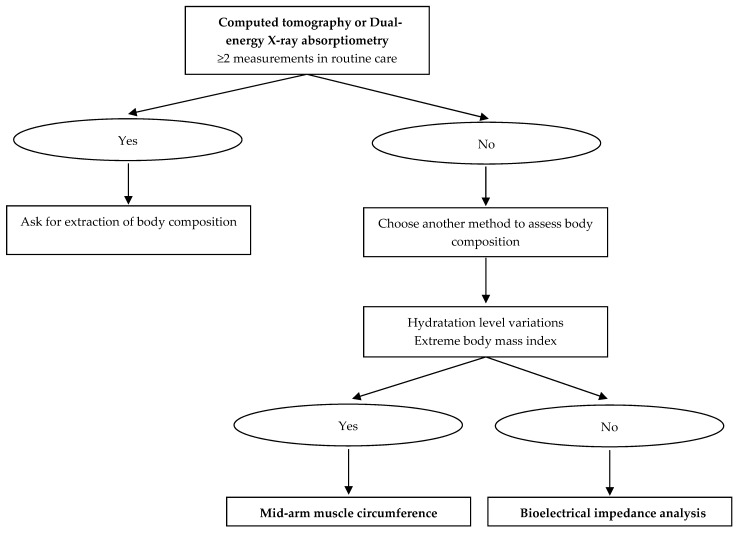
Muscle mass assessment in clinical practice: which method?

**Table 1 jcm-08-01040-t001:** Principal characteristics of main clinical methods to assess muscle mass.

	MAMC	BIA	DXA	CT
Accuracy	-	+	++	+++
Interobserver variability	+++	+	-	-
Simplicity	++	++	+	-
Radiation	-	-	+	+++
Cost				
*If device already available*	-	-	- ^1^	+ ^2^
*If device not available*	-	+	++	+++
Time to measurement	5 min	5 min	5–10 min ^3^	10–15 min ^3^

“-“: weak/low; “+“: high; ^1^ Body composition software usually included in the device; ^2^ Related to the purchase of the software; ^3^ To obtain body composition analysis in addition to a routine exam; MAMC: mid-arm muscle circumference; BIA: bioelectrical impedance analysis; DXA: dual-energy X-ray absorptiometry; CT: computed tomography; Adapted from Guglielmi and al. [11].

**Table 2 jcm-08-01040-t002:** Randomized controlled trials: effects of nutritional or physical interventions on muscle mass in patients with cancer.

	Studies	Population	Intervention Group	Comparison Group	Muscle Mass	Significant Results
**Nutrition**	Ritch et al. 2019 [91]	Urothelial bladder carcinoma undergoing radical cystectomy INT = 31/CO = 30	Daily oral nutritional supplement with ω-3 and HMB(700 kcal, 26 g proteins) 4 weeks before and after surgery	Oral micronutrients 2×/day	CT	30 days post-operatively: - ↓ patients with SMI loss - No impact on hospital length of stay, postoperative complications, readmissions and mortality
	Burden et al. 2017 [92]	Colorectal cancer INT = 55/CO = 46	Daily oral nutritional supplement (600 kcal, 24 g proteins) ≥5 days before surgery+ dietary advice	Dietary advice	BIA	5–7 days post-operatively: - No impact on FFMI and postoperative complications - ↓ % weight loss and surgical site infection
**Physical exercise**	Galvao et al. 2018 [93]	Metastatic prostate cancer INT = 28/CO = 29	Supervised endurance, resistance and flexibility exercises 3 months, 3×/week, 60 min	Usual physical activity	DXA	After 3-month intervention: - No impact on lean soft tissue - ↑ self reporting physical functioning and leg strength
	Taaffe et al. 2018 [94]	Prostate cancer with previous androgen deprivation therapy and radiotherapy INT = 50/CO = 50	Supervised endurance and resistance exercises 6 months, 2×/week, 60 min followed by home-based endurance, resistance and flexibility exercises 6 months, 2×/week, 60 min	Recommendation for 150 min/week of moderate intensity physical exercise for 12 months based on educational material	DXA	After 6-month intervention: - ↑ ASMM, chair rise time, leg and arm strength - No impact on for lean soft tissue After 12-month intervention: - No impact on ASMM, leg strength and lean soft tissue - ↑ chair rise time and arm strength
	Wall et al. 2017 [32]	Prostate cancer undergoing androgen deprivation therapy INT = 60/CO = 47	Supervised endurance and resistance exercises 6 months, 2×/week, 60 min + home-based endurance exercise 6 months, 150 min/week	Usual physical activity	DXA	After 6-month intervention: - ↑ lean soft tissue, V0_2max_, fat oxidation - No impact on resting metabolic rate, carbohydrate oxidation and body weight
	Adams et al.2016 [95]	Breast cancer undergoing adjuvant chemotherapy INT endurance = 66 INT resistance = 64 CO = 70	During chemotherapy: INT endurance 3×/week, 105 min INT resistance 3×/week	Usual physical activity	DXA	At the end of chemotherapy: INT resistance VS CO: - ↑ lean soft tissue index, leg and arm strengthINT endurance VS CO: - No impact on lean soft tissue, leg and arm strength INT resistance VS INT endurance:- No impact on lean soft tissue - ↑ leg and arm strength

INT: intervention group, CO: control group, ω-3: omega-3 fatty acids, HMB: β-hydroxy β-methyl butyrate, CT: computed tomography, BIA: bioelectrical impedance analysis, DXA: dual-energy X-ray absorptiometry, SMI: skeletal muscle index, FFMI: fat-free mass index, ASMM: appendicular skeletal muscle mass.

**Table 3 jcm-08-01040-t003:** Randomized controlled trials (>100 participants): effects of nutritional or combined nutritional and physical interventions on muscle mass in older adults.

	Studies	Population	Intervention Group	Comparison Group	Muscle Mass	Significant Results
**Nutrition**	Cramer et al. 2016 [96]	Malnutrition and sarcopenia in the community INT = 165/CO = 165	Daily oral nutritional supplement with HMB (660 kcal, 40 g proteins) + Usual diet during 24 weeks	Daily oral nutritional supplement (660 kcal, 28 g proteins) + Usual diet	DXA	After 24-week intervention, in both groups: - No impact on lean soft tissue - ↑ FM, handgrip strength, gait speed, muscle quality and isokenetic peak torque leg strength - No outcome difference between groups
	Malafarina et al. 2017 [97]	Traumatic hip fracture in rehabilitation hospital INT = 55/CO = 52	Daily oral nutritional supplement with HMB (660 kcal, 40 g proteins) + Standard diet 1500 kcal, 87.4 g protein during rehabilitation stay	Standard diet 1500 kcal, 87.4 g protein	BIA	At the end of the rehabilitation: - ↓ FFM, ASMM and BMI decrease - No impact on handgrip strength, gait speed
**Nutrition and physical exercise**	Englund et al. 2017 [98] Fielding et al. 2017 [99]	Mobility-limitation and vitamin D insufficiency in the community INT = 74/CO = 75	Daily oral nutritional supplement (150 kcal, 20 g whey protein, 800UI vit D) + Supervised endurance, resistance, balance and flexibility exercises 3×/week, 60 min during 6 months	Daily placebo (30 kcal) + Supervised endurance, resistance, balance and flexibility exercises 3×/week, 60 min during 6 months	DXA	After 6-month intervention, in both groups: - No impact on ASMM - ↑ muscle strength, thigh muscle composition, gait speed, short physical performance battery score - ↓ FM - ↑ lean soft tissue only for control group - ↑ 25(OH)D only for intervention group - No outcome difference between groups

INT: Intervention group, CO: Control group, HMB: β-hydroxy β-methyl butyrate, DXA: dual-energy X-ray absorptiometry, BIA: bioelectrical impedance analysis, FM: fat mass, FFM: fat-free mass, ASMM: appendicular skeletal muscle mass, BMI: body mass index.

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
