# Peer review of "Clinical Value of Muscle Mass Assessment in Clinical Conditions Associated with Malnutrition"

_jcm, 2019, doi:10.3390/jcm8071040_

Reviewer 1 Report

I thoroughly enjoyed reading this manuscript on a relevant and clinically important topic. This article has the potential to impact readers in a positive direction.

My personal favorite line (268-269): "Of note, being able to derive body composition calculation from pre-existing clinical images is of great interest from both clinical and economic perspective."

However, significant flaws exist that need to be addressed for publication in this well-cited journal. The title suggests that the authors believe there is clinical utility in assessing body composition. But, to convince readers of the clinical utility of body composition, the authors must overcome the obvious question, “How does body composition assessment improve upon a simple BMI calculation?” (See comments below)

I hope the authors find the additional comments below constructive and I look forward to reviewing any future copies of this manuscript.

Major Comments:

1.       The paper is described as a "narrative review," while it reads more similarly to a systematic review. For a systematic review, I am sure the authors are aware, the methods are not rigorous enough. However, as a narrative review, describing the search methods are not necessary and given the weak and limited search criterion and database (singular!) used, this information lessens my perception of the manuscript. Consider removing these descriptions as it does not add to the quality of the final product. Alternatively, consider the opposite direction and bolster the search criterion and descriptions, for example, (see major comment #3) describe why studies using less than 100 subjects were excluded from analysis to show the thought process behind a seemingly arbitrary cutoff. Going either route – removing the descriptions completely, or enhancing them to the point they significantly add to the final product will improve the manuscript compared to the current form.

2.       Figure 1 is unclear. It reads as if muscle mass reduces physical function, increases chemotherapy toxicity, reduces quality of life etc. Please rework.

3.       Figure 2 and related information: Why were studies only included if they had >100 subjects? This seems arbitrary and raises a concern of author bias. How many studies were excluded from the review for having<100 subjects? Is it 1 or 100 studies?

4.       Figure 3 requires additional commentary on the clinical application. As the review suggests, body composition may have clinical importance. However, from the examples provided in Figure 3, I cannot see any potential examples of “cutoffs” or clinically relevant identification points that would be useful in treatment. Observing reductions in fat-free mass are not unusual, and often physically visible and detectable by simple BMI calculations – how will the enhanced granularity of body composition assessment improve care/outcomes? Consider adding specific patient events tied with changes in body composition on the timelines for the figures – for example: a cardiac event, long hospital stay, documented edema, etc. This may reveal to readers how they could use potentially use this enhanced body composition assessment in a timely manner, before changes in BMI may become present.

5.       To make the primary thrust of his review clinically relevant, more discussion on why body composition assessment is better than a simple BMI calculation is needed. The ease, simplicity and long-standing effectiveness of the BMI makes it difficult to justify the clinical application of the increased technicality and time required for the methods discussed within. I agree fully with the authors’ opinion on the clinical value of body composition assessments, however, if I remove my bias, the paper in its current form does not convince me of its potential benefit. (see points 4 and 6 for options)

6.       Muscle function tests (grip strength, sit-stand time, etc) are associated with patient outcomes and mortality in community dwelling older adults – however, these are difficult to perform from both the practitioner and patient perspective. A great surrogate is body composition. Perhaps the article would benefit from discussing the relationship of body composition and muscle function tests, evidencing a benefit beyond BMI.

7.       More references and discussion of the clinical application of body composition measures are needed – currently it reads as an “author opinion” but there is plenty of data to support this as well. Here is an example: https://www.ncbi.nlm.nih.gov/pubmed/31192685.

 Minor Comments:

1.       Generally, the paper requires additional proofreading.

2.       GOLD and BODE indices are not defined.

3.       Several instances of "single-sentence paragraphs." These are awkward for the reader, please remove by merging single-sentence paragraphs within the previous paragraph (where appropriate).

4.       Typo: line 145

5.       Line 206-207 is not generally accepted phrasing. Consider "as described above" or rework the sentence. Also, this is an example of a "single sentence paragraph."

6.       Line 214 "both" should be replaced with "either"

7.       Line 238-239 The use of the RCT abbreviation reads awkwardly. Regardless, there is not a need to abbreviate as "RCT," it would be easier on the reader to spell out "randomized controlled trials" in the few instances of use (only used 3 times, no need for an abbreviation that may disrupt readability)

Stylistic Comments:

1.       Line 247 – this is not common terminology. Consider “older adults” or “aging” – also, the word “patients” can be removed from each of the subheadings between line 208 and 247.

2.       Consider adjusting line (268-269): "Of note, being able to derive body composition calculation from pre-existing clinical images is of great interest from both clinical and economic perspective." to: "Of note, deriving body composition from pre-existing clinical images is an opportunity to improve diagnosis and treatment without additional cost or patient burden. Clinical research in this area is warranted."

3.       There may be an opportunity to discuss the application of body composition measures to improve interdisciplinary care (https://journals.lww.com/topicsingeriatricrehabilitation/Abstract/2019/01000/The_Crossroads_of_Aging__An_Intersection_of.8.aspx) – may be another option to address Major Comment #5.

4.       The conclusion paragraph is weak. Please rework

Author Response

Manuscript ID: jcm-526971

Type of manuscript: Review

 Title: Clinical value of muscle mass assessment in clinical conditions associated with malnutrition

Authors: Julie Mareschal*, Najate Achamrah, Kristina Norman, Laurence Genton

 For the answer, page numbers and line numbers refer to the document:
"
jcm-526971 ori_Manuscript revision Track changes accepted_2019 06 27"

 Answers to reviewer 1

I thoroughly enjoyed reading this manuscript on a relevant and clinically important topic. This article has the potential to impact readers in a positive direction.

My personal favorite line (268-269): "Of note, being able to derive body composition calculation from pre-existing clinical images is of great interest from both clinical and economic perspective."

However, significant flaws exist that need to be addressed for publication in this well-cited journal. The title suggests that the authors believe there is clinical utility in assessing body composition. But, to convince readers of the clinical utility of body composition, the authors must overcome the obvious question, “How does body composition assessment improve upon a simple BMI calculation?” (See comments below)

I hope the authors find the additional comments below constructive and I look forward to reviewing any future copies of this manuscript.

Thank you very much for these encouraging comments. Please find below the point-by-point detailed response to the questions that were raised.

Major Comments:

1.       The paper is described as a "narrative review," while it reads more similarly to a systematic review. For a systematic review, I am sure the authors are aware, the methods are not rigorous enough. However, as a narrative review, describing the search methods are not necessary and given the weak and limited search criterion and database (singular!) used, this information lessens my perception of the manuscript. Consider removing these descriptions as it does not add to the quality of the final product. Alternatively, consider the opposite direction and bolster the search criterion and descriptions, for example, (see major comment #3) describe why studies using less than 100 subjects were excluded from analysis to show the thought process behind a seemingly arbitrary cutoff. Going either route – removing the descriptions completely, or enhancing them to the point they significantly add to the final product will improve the manuscript compared to the current form.

We thank the reviewer for this relevant comment with which we agree. We decided to remove the descriptions and only mention on page 4, line 144 “We made an update of the latest observational studies and subjectively considered articles from the last three years”. Indeed, as mentioned in the introduction, this review is a narrative review.

 2.       Figure 1 is unclear. It reads as if muscle mass reduces physical function, increases chemotherapy toxicity, reduces quality of life etc. Please rework.

To clarify the message of the figure 1, we specified that a low muscle mass is associated with all of these different conditions.

 3.       Figure 2 and related information: Why were studies only included if they had >100 subjects? This seems arbitrary and raises a concern of author bias. How many studies were excluded from the review for having<100 subjects? Is it 1 or 100 studies?

In figure 2 and table 3, we actually considered studies > 100 subjects. Indeed, over the last three years we found about 80 articles dealing with muscle mass in the older adults. Most of the studies having > 100 subjects gave the same results as those with fewer subjects. As we were limited in the number of references, we decided to select only studies having > 100 subjects. To clarify our choice we added on page 7, line 194 “Over the last three years, we found over 80 studies dealing with muscle mass in the older adults. Due to the large amount of publications, figure 2 considers studies including more than 100 subjects and illustrates the effect of low muscle mass according to the clinical setting. Studies including fewer subjects showed a similar impact.” and page 10, line 252 “To limit the size of the table and facilitate the reading, we reported studies including over 100 subjects, but the results were similar in studies with fewer subjects.”

 4.       Figure 3 requires additional commentary on the clinical application. As the review suggests, body composition may have clinical importance. However, from the examples provided in Figure 3, I cannot see any potential examples of “cutoffs” or clinically relevant identification points that would be useful in treatment. Observing reductions in fat-free mass are not unusual, and often physically visible and detectable by simple BMI calculations – how will the enhanced granularity of body composition assessment improve care/outcomes? Consider adding specific patient events tied with changes in body composition on the timelines for the figures – for example: a cardiac event, long hospital stay, documented edema, etc. This may reveal to readers how they could use potentially use this enhanced body composition assessment in a timely manner, before changes in BMI may become present.

We agree with the reviewer that this information is important. As suggested, we added specific patient events in the figure legend. Moreover, we discussed these examples in the text page 12, line 270“Examples of patients sample are presented in figure 3. They highlight the clinical importance of body composition assessment to detect changes in muscle mass according to every specific patient event. Furthermore, these examples illustrate the advantage of body composition over BMI. For example, figure 3a shows a stabilization of FFM but a decrease of total body weight and thus of BMI between July and September 2018. In figure 3c, BMI tends to decrease as MAMC increases.”

 5.       To make the primary thrust of his review clinically relevant, more discussion on why body composition assessment is better than a simple BMI calculation is needed. The ease, simplicity and long-standing effectiveness of the BMI makes it difficult to justify the clinical application of the increased technicality and time required for the methods discussed within. I agree fully with the authors’ opinion on the clinical value of body composition assessments, however, if I remove my bias, the paper in its current form does not convince me of its potential benefit. (see points 4 and 6 for options)

We thank the reviewer for this relevant comment. To improve the manuscript, we added on page 2, line 52: “In clinical practice, several tools and techniques are available to assess body composition. BMI is often used due to its simplicity. Indeed, the U-shaped association between BMI and all-cause mortality has been well described: subjects with the highest and lowest BMIs have the highest mortality rates [5]. However, BMI does not allow to measure body composition compartments and tends to underestimate fat-free mass (FFM) depletion [6-8]. “ and on page 12, line 261: “Although convenient and quick, BMI has shown limitations in the screening and the follow-up of malnutrition with a tendency to underestimate muscle mass [6-8]. The disparity between BMI and FFM raise the need for a precise quantitative evaluation of muscle mass or muscle function to both direct and validate the effects of clinical interventions in malnourished patients.

 6.       Muscle function tests (grip strength, sit-stand time, etc) are associated with patient outcomes and mortality in community dwelling older adults – however, these are difficult to perform from both the practitioner and patient perspective. A great surrogate is body composition. Perhaps the article would benefit from discussing the relationship of body composition and muscle function tests, evidencing a benefit beyond BMI.

The primary aim of this review is to deal with muscle mass but not muscle strength or physical performance. However, we agree with the reviewer that muscle function tests should be considered as part of diagnostic criteria of sarcopenia (ref: Cruz A, Age Ageing, 2018). We added on page 12, line 261: “Although convenient and quick, BMI has shown limitations in the screening and the follow-up of malnutrition with a tendency to underestimate muscle mass [6-8]. The disparity between BMI and FFM raise the need for a precise quantitative evaluation of muscle mass or muscle function to both direct and validate the effects of clinical interventions in malnourished patients. Indeed, it has been established that a loss of muscle mass is associated with a decrease in physical function or muscle strength [101].”

7.       More references and discussion of the clinical application of body composition measures are needed – currently it reads as an “author opinion” but there is plenty of data to support this as well. Here is an example: https://www.ncbi.nlm.nih.gov/pubmed/31192685.

As suggested for points 4, 5, and 6, we improved the discussion of the clinical application of body composition through the manuscript. Moreover, we added references to support your opinion in paragraph 5.

Minor Comments:

1.       Generally, the paper requires additional proofreading.

As required, we did additional proofreading of the manuscript.

 2.       GOLD and BODE indices are not defined.

Page 4, line 153, we defined GOLD and BODE indices “In this study, disease severity was evaluated with the Global Initiative for Obstructive Lung Disease (GOLD) index based on airflow limitation, exacerbation history and symptom burden, and prognosis with the Body mass index/Airflow obstruction/Dyspnea/Exercise capacity (BODE) score, known to predict disease severity and mortality”.

 3.       Several instances of "single-sentence paragraphs." These are awkward for the reader, please remove by merging single-sentence paragraphs within the previous paragraph (where appropriate).

This was change as requested where appropriate (e.g. page 4, line 165).

 4.       Typo: line 145

We reworked the typography of all the headings and subheadings.

 5.       Line 206-207 is not generally accepted phrasing. Consider "as described above" or rework the sentence. Also, this is an example of a "single sentence paragraph."

In response to the suggestion of the reviewer, we changed the sentence, page 8, line 209 to “In this section, we focused on non-pilot randomized controlled trials published during the last 3 years. We did not find new relevant studies with growth hormone supplementation.

 6.       Line 214 "both" should be replaced with "either"

As requested, “both” was changed to “either”.

7.       Line 238-239 The use of the RCT abbreviation reads awkwardly. Regardless, there is not a need to abbreviate as "RCT," it would be easier on the reader to spell out "randomized controlled trials" in the few instances of use (only used 3 times, no need for an abbreviation that may disrupt readability)
The change was made as suggested in the manuscript on page 8, line 209.

Stylistic Comments:

1.       Line 247 – this is not common terminology. Consider “older adults” or “aging” – also, the word “patients” can be removed from each of the subheadings between line 208 and 247.

Thank you for the recommendation. We changed “old patients” to “older adults” in the manuscript. We also removed the word “patients” in all the subheadings between line 147 and 239.

 2.       Consider adjusting line (268-269): "Of note, being able to derive body composition calculation from pre-existing clinical images is of great interest from both clinical and economic perspective." to: "Of note, deriving body composition from pre-existing clinical images is an opportunity to improve diagnosis and treatment without additional cost or patient burden. Clinical research in this area is warranted."

The change was made as suggested.

3.       There may be an opportunity to discuss the application of body composition measures to improve interdisciplinary care (https://journals.lww.com/topicsingeriatricrehabilitation/Abstract/2019/01000/The_Crossroads_of_Aging__An_Intersection_of.8.aspx) – may be another option to address Major Comment #5.

In response to the suggestion of the reviewer, we added “Finally, train clinicians to routinely use body composition in their practice and interpret the results should be included in pre- and post-graduated educational programs, as proposed by the European Society for Clinical Nutrition and Metabolism Life Long Learning (LLL) program. The awareness and the training of clinicians to body composition assessment may be a great opportunity to improve interdisciplinary care in the screening and management of malnutrition “on page 15, line 299.

4.       The conclusion paragraph is weak. Please rework

As request by the reviewer, we reworked the conclusion “Available data suggest that precise assessment of body composition might be clinically relevant in the management of malnourished patients. Various methods have been validated to measure muscle mass. The method selection should be driven by the clinical situation, the patient’s characteristics, and logistic and economic parameters. Standard measures of body composition such as BMI are valuable for their simplicity in daily practice. They do however not reflect body composition compartments. There is growing evidence in the litterature that repartition of muscle mass, is a more valuable tool in this assessment. Knowing this repartition allows a tailored approach to the nutritional treatment of malnourished patients and, according to the literature, led to improved clinical outcomes in various chronic diseases as well as in the older adult. Therefore, there is a need to more systematized data to orientate upcoming clinical guidelines of body composition assessment in malnourished patients.”

Reviewer 2 Report

The review should indicate the position in which MAMC and other measures are obtained (eg sitting, supine).

The authors should note that the presence of metal may affect BIA.

It’s best to use “people first language (eg, Patients with chronic obstructive pulmonary disease).

The authors mention the use of anabolic steroids for increasing muscle mass. What about growth hormone.

I expected more on the role of exercise on muscle mass.

For references 9,23,28,35,43,55,56,59,61,73,85,86,88,91,93,94,95 the article title is inappropriately capitalized.

Not exactly sure why the specific diseases are chosen. Diabetes and peripheral arterial disease also affect muscle mass.

Author Response

Manuscript ID: jcm-526971

Type of manuscript: Review

 Title: Clinical value of muscle mass assessment in clinical conditions associated with malnutrition

Authors: Julie Mareschal*, Najate Achamrah, Kristina Norman, Laurence Genton

 For the answer, page numbers and line numbers refer to the document:
"
jcm-526971 ori_Manuscript revision Track changes accepted_2019 06 27"

 Answers to reviewer 2

We are grateful for the useful comments on our manuscript.

Please find below the point-by-point detailed response to the questions that were raised.

 1.       The review should indicate the position in which MAMC and other measures are obtained (eg sitting, supine).

As suggested by the reviewer, we mentioned the position in which MAMC, BIA, DXA and CT are performed.

For MAMC: Page 2, line 68:  “Measurements are usually performed in standing position, on the dominant arm, at the mid-point between the acromion and the olecranon.”

For BIA: Page 3, line 87:  “Briefly, the patient, lying on the back, is exposed to a low intensity alternating current between surface electrodes.”

For DXA: Page 3, line 113: “Measurement is achieved in supine position.”

For CT: Page 3, line 126: “CT is a medical imaging technique, performed in supine position, which measures tissue absorption of x-rays emitted through a rotating beam.”

2.       The authors should note that the presence of metal may affect BIA.

To our knowledge, BIA measurements are not influenced by metal or implantable cardioverter-defibrillators (ref: Mehlig K, Clin Nutr ESPEN, 2015 and  Meyer P, J Parenter Enteral Nutr, 2017). Therefore, we did not mention in the review that metal may affect BIA.

 3.       It’s best to use “people first language (eg, Patients with chronic obstructive pulmonary disease).

Thank for your suggestion. We changed titles and subtitles. We also made some changes in the text (e.g. Page 4, line 152).

4.       The authors mention the use of anabolic steroids for increasing muscle mass. What about growth hormone.

We agree that some studies showed that growth hormone may improve muscle mass in specific population. For example, in older adults (ref: Liu H, Ann Intern Med, 2007) or in patients with chronic obstructive pulmonary disease (ref: Burdet L, Am J Respir Crit Care Med, 1997).  However, as mentioned in the manuscript, we selected only studies from the last three years and we did not find new interventional studies studying the impact of growth hormone supplementation on muscle mass. We added the following sentence on Page 8, line 206: “Anabolic steroids and growth hormone have been considered in malnourished patients but no clinical practice guideline has yet been published [81, 82]. […]  In this section, we focused on non-pilot randomized controlled trials published during the last 3 years. We did not find new relevant studies with growth hormone supplementation.”

5.       I expected more on the role of exercise on muscle mass.

Indeed, the role of exercise on muscle mass is very important. However, this was not the aim of the review and would likely require a review on its own. Based on the literature of the last three years, we already cited all pertinent randomized control trials with physical exercise interventions for each selected specific diseases. Moreover, on page 8, line 203, we mentioned “Regular physical exercise is promoted for patients with COPD, chronic heart failure and cancer as it improves cardiorespiratory fitness, muscle mass and strength, quality of life, and decrease COPD exacerbation and chemotherapy toxicity”.

To conclude, we think that physical exercise, nutrition and anabolic steroids are all strategies to improve muscle mass. In each section of paragraph 4, we mentioned the beneficial effects of each intervention, focusing on the publications of the last 3 years, but this review did not aim to detail these modalities.

6.       For references 9,23,28,35,43,55,56,59,61,73,85,86,88,91,93,94,95 the article title is inappropriately capitalized.

For all these references, we corrected the title by replacing capital letters with lower case letters.

7.       Not exactly sure why the specific diseases are chosen. Diabetes and peripheral arterial disease also affect muscle mass.

We had to focus on some diseases, as reviewing all diseases is not realistic for a review. As mentioned, we choose diseases with the highest prevalence of malnutrition (ref: Von Haehling S, J Cachexia Sarcopenia Muscle, 2016).  Diabetes and peripheral arterial diseases are not among them. We mentioned on page 4, line 142: “We focused on chronic diseases such as chronic obstructive pulmonary disease (COPD), chronic heart failure (CHF), and cancer and on the older adults. This choice was related to the fact that the prevalence of malnutrition is particularly high in these populations“.

Round  2

Reviewer 1 Report

Exceptional responses and manuscript edits with a quick turnaround!

Author Response

Thank you very much for this nice comment !

Reviewer 2 Report

Some awkward language exists. A few examples follow. Replace “impact” (line 17) with “impacts.” Replace “This review aims at” with “This review aims to’ (lines 18 & 48). Replace ‘to evaluate muscle mass for screening” with “of assessing  muscle mass as part of the screening…”(line 21). Replace “does not allow to measure” with “does not allow the measurement” (line 55). Eliminate “measurements” (line 57), it is redundant. Replace “but allowing simultaneously a body composition assessment” with “but allows for the simultaneous assessment of body composition” (line 63).

Nowadays the term “participants” is considered more appropriate than “subjects.”

Author Response

Thank for your suggestions to improve the manuscript. Please find below the point-by-point detailed response to the suggestions that were raised.

Some awkward language exists. A few examples follow.

· Replace “impact” (line 17) with “impacts.” This was change as requested line 17.

· Replace “This review aims at” with “This review aims to’ (lines 18 & 48). This was change as requested line 18 & 47.

· Replace ‘to evaluate muscle mass for screening” with “of assessing  muscle mass as part of the screening…”(line 21). This was change as requested line 21.

· Replace “does not allow to measure” with “does not allow the measurement” (line 55). This was change as requested line 54.

· Eliminate “measurements” (line 57), it is redundant. This was change as requested line 56.

· Replace “but allowing simultaneously a body composition assessment” with “but allows for the simultaneous assessment of body composition” (line 63). This was change as requested line 62.

· Nowadays the term “participants” is considered more appropriate than “subjects.” As suggested by the reviewer, we replaced the term “subjects” with the term “participants” (e.g. lines 190, 191, 245, 246).